# What Does It Take to Play the Piano? Cognito-Motor Functions Underlying Motor Learning in Older Adults

**DOI:** 10.3390/brainsci14040405

**Published:** 2024-04-20

**Authors:** Florian Worschech, Edoardo Passarotto, Hannah Losch, Takanori Oku, André Lee, Eckart Altenmüller

**Affiliations:** 1Institute of Music Physiology and Musician’s Medicine, Hanover University of Music, Drama and Media, 30175 Hanover, Germany; 2Center for Systems Neuroscience, 30559 Hanover, Germany; 3Department of Neuroscience, University of Padova, 35121 Padova, Italy; 4Institute for Music Education Research, Hanover University of Music, Drama and Media, 30175 Hanover, Germany; 5NeuroPiano Institute, Kyoto 600-8086, Japan; 6College of Engineering and Design, Shibaura Institute of Technology, Tokyo 135-8548, Japan; 7 Department of Neurology, Klinikum Rechts der Isar Technische Universität München, 80333 Munich, Germany

**Keywords:** motor sequence learning, playing music, skill, cognition, performance–ability relationship, working memory, processing speed, transfer

## Abstract

The acquisition of skills, such as learning to play a musical instrument, involves various phases that make specific demands on the learner. Knowledge of the cognitive and motor contributions during learning phases can be helpful in developing effective and targeted interventions for healthy aging. Eighty-six healthy older participants underwent an extensive cognitive, motoric, and musical test battery. Within one session, one piano-related and one music-independent movement sequence were both learned. We tested the associations between skill performance and cognito-motor abilities with Bayesian mixed models accounting for individual learning rates. Results showed that performance was positively associated with all cognito-motor abilities. Learning a piano-related task was characterized by relatively strong initial associations between performance and abilities. These associations then weakened considerably before increasing exponentially from the second trial onwards, approaching a plateau. Similar performance–ability relationships were detected in the course of learning a music-unrelated motor task. Positive performance–ability associations emphasize the potential of learning new skills to produce positive cognitive and motor transfer effects. Consistent high-performance tasks that demand maximum effort from the participants could be very effective. However, interventions should be sufficiently long so that the transfer potential can be fully exploited.

## 1. Introduction

When studying motor behavior, two types of motor learning are commonly differentiated: *motor sequence learning*, defined as the acquisition of movements into a well-executed behavior; and *motor adaption*, which is the ability to successfully execute an already well-practiced movement in changing environments [1,2]. Learning to play the piano is a complex form of motor skill acquisition. Although making music also relates to motor adaptation (e.g., compensating for changing muscle properties due to fatigue; see [3]), the process of learning a musical instrument or acquiring a new musical repertoire is better captured by motor sequence learning models. This distinction is crucial as cortico-striatal, cortico-cerebellar, and limbic systems, as well as the spinal cord, are supposed to contribute in different ways to both types of motor learning and their different phases [4,5]. Reybrouck and Schiavio [6] provided an informative overview of the main theories and concepts related to skill acquisition and motor learning, with a particular focus on musical performance.

### 1.1. Cognitive Contributions of Motor Sequence Learning

The acquisition of skills is believed to undergo separate stages featuring parallel and interacting processes [4,7,8,9,10], beginning with an initial fast learning stage with rapid improvements and reaching a final stage achieving individual asymptotic performance. Although very differently labeled (cognitive–associative–autonomous [8]; phase 1–phase 2–phase 3 [7]; knowledge-based–rule-based–skill-based [11]; declarative–declarative/procedural–procedural [12]; exploration–selection–refinement [13]; early–middle–late [14]; etc.), there is a general agreement on a three-stage process of learning, with a few exceptions (e.g., four and five stages in the learning models of Verwey [15] and Doyon and Benali [1], respectively). Ackerman [7] predicted that the performance of each motor learning stage is determined by a different set of abilities. An individual’s general intelligence and task-specific knowledge are suggested to be particularly important in the early stage. After initial practice, perceptual speed becomes the dominant factor impacting performance. Finally, as consolidation and automatization progress—and if the task is *consistent*—general intelligence and perceptual speed become less important, and psychomotor abilities increasingly contribute to performance [7,16]. 

Task consistency is a very important moderator of the relationship between ability and performance. Due to fixed stimulus–response mapping, it is suggested that consistent tasks can be accomplished automatically or at least with a reduced allocation of cognitive resources with increased practice. The Challenge point framework [17] provides a useful illustration of this as it assumes that a subjectively simple task is associated with a relatively low amount of information that can or must be processed. A balanced relationship between task demands and individual capabilities often induces experiences of *flow*, a state of mind that is perceived as “automatic” and “effortless” [18]. It is a phenomenon frequently described by musicians when performing well-rehearsed pieces of music with a manageable level of difficulty [19,20]. Inconsistent tasks, on the other hand, are characterized by changing stimulus–response pairings and thus changing demands on information processing. These tasks must still be cognitively controlled and therefore remain cognitively demanding [21,22,23]. Inconsistent tasks can be adaptive tasks in which the difficulty and stimulus–response mappings change according to the participant’s performance. However, even well-practiced motor tasks can require continuous controlled processing despite a fixed stimulus–response mapping [24]. For example, manual gear shifting appears to be a controlled process well beyond the first year of driving a car [25]. Similar results were found for rowing, which requires cognitive monitoring not only for beginners but also for top athletes with more than 10 years of training [26,27]. Further, the execution of simple discrete finger sequences practiced several hundred times cannot be regarded as automatic [15]. Even the speed of walking, the most basic form of human gait with approximately 300 million steps in the lifetime of an 80-year-old (assuming s/he walked the commonly recommended 10,000 steps per day), is positively associated with cognition in older adults [28]. 

In the following, we refer to consistent tasks that continuously demand maximum effort from participants as *consistent high-performance tasks*. These are tasks with high functional (i.e., subjective) task difficulty, in which participants, for example, have to perform an identical task as quickly and accurately as possible over several rounds.

Beaunieux et al. [29] tested performance–ability relationships using a consistent high-performance task following a massed learning paradigm. Over 40 trials, 100 younger participants learned a procedural task: the Tower of Toronto (This test contains three rods and a stack consisting of a rectangular base and four differently colored discs. The task is to rebuild the stack from the left to the right rod, obeying the following two rules: Only one disc may be moved at a time, and a dark-colored disc may never be placed on top of a light-colored disc.) [30]. Over the 40 trials, the authors showed decreasing correlations between performance and both fluid intelligence and episodic memory function but an increasing correlation between psychomotor skills and performance. These results were replicated in a second experiment in which older participants practiced the Tower of Toronto over four consecutive days [29]. The correlation between performance and perceptual speed remained relatively stable throughout the learning process and even appeared to increase. Growing performance–ability associations were also the result of a cross-sectional study including a U.S. national cohort of 4,126 pupils from ages 6 to 19 [31]. The authors showed an increasing influence of processing speed on reading achievement over the course of schooling. 

Working memory is usually assumed to be associated with early performance [9,29,32,33]. However, an increasing working memory demand during practice is also conceivable due to growing task complexity. This was demonstrated in reading and writing achievement [31,34].

### 1.2. Playing Music as a Special Type of Motor Sequence Learning

Playing and learning a musical instrument can be considered a special form of motor sequence learning. Abstract symbols on the music sheet have to be encoded and translated into a motor response, taking into account auditory, somatosensory, and visual feedback [6,35]. The motor response involves the execution of often very fast and quite complex movements with millisecond precision [36,37], adhering to both the correct sequence of notes and specified volume. The high processing requirements associated with this complexity motivated many scientists to investigate the cognitive effects of music-making [33,38,39,40]. However, only very few studies linked individual cognitive baseline profiles to individual learning outcomes. Burgoyne et al. [41] showed that general intelligence—including measures of working memory and processing speed—accounted for 21.4% of the variance in a novel piano task. As the performance was measured only three times (before, during, and after) in a single session, individual learning trajectories and the stage of learning that was reached by the participant could not be adequately assessed. In other words, the stage of learning in which general intelligence supports (piano-related) motor performance could not be determined. This problem was avoided by Kopiez and Lee [42], who described performance–ability relationships during a clearly delimited learning phase. They asked 52 piano majors, graduates, and postgraduates to *sight read*, a very basic musical skill that is required of all professional musicians. When sight reading, which is the unrehearsed performance of music and an excellent example of the earliest motor learning phase, the authors demonstrated that performance correlated with processing speed [r ranging from 0.27 to 0.51], working memory [0.08, 0.32], Short-term memory [0.04, 0.3] and psychomotor speed [0.05, 0.27]. However, conclusions about the dynamics of performance–ability relationships throughout the learning process could also not be made.

### 1.3. Transfer of Music Training beyond the Musical Domain

Beneficial cognitive effects induced by motor–cognitive training like video gaming [43,44], Tai Chi [45], or coordination exercises [46] are frequently reported. This has led the World Health Organization [47] to recommend cognitive training and physical activity to reduce cognitive decline and dementia. Due to the reasons mentioned above, it is reasonable to assume that also music training can induce effects beyond the musical domain. Yet, meta-analyses examining the transferability of music making to cognitive domains yield conflicting results [48,49,50,51], and some researchers have raised doubts about the possibility of transfer in general [52]. 

In intervention studies, null effects are often explained by short intervention durations, leaving insufficient time for adaptation and thus hindering transfer. However, another important condition must be met so that transfer can take place. According to Thorndike and Woodworth [53], transfer necessitates sufficient similarity with shared common elements between the trained and the transfer domain. The stronger the correlation between two tests, the higher the probability of transfer from one to the other. This conception still holds and is part of modern theories of transfer [54]. Moreover, the associations between training (e.g., piano performance) and transfer domain (e.g., working memory) may change over the learning period, leading to specific “sensitive learning phases” for each cognitive domain. Based on this, we argue that different study results do not necessarily contradict each other but are reflected as a consequence of changing performance–ability relationships over the course of the learning process. In addition, the strength of these associations indicates the “overlap” between the trained and transfer areas on a continuous scale. This can provide information about the likelihood and potential of a transfer and has several advantages over the common distinction between “near” and “far” transfer.

### 1.4. Aim of the Study

The aim of this project was to lay the foundation for the development of a comprehensive cognitive process theory of music making. Within this theory, cognitive and motor functions underlying the different phases of motor sequence learning were to be identified. Specifically, our research aimed to investigate whether working memory, processing speed, psychomotor speed, and dexterity are related to piano performance and whether these relationships between performance and abilities change during the learning process. In this study, we used piano performance to operationalize motor behavior during the learning process of an unknown musical piece. The task was designed at an appropriate level of difficulty so that the performance across trials reflects a maximum learning process, albeit within a single session. It should be noted, however, that our study can only consider a section from a developing learning process and thus cannot conclusively clarify whether this section is limited to “early” learning or already includes phases of “later” learning—after all, the training period lasted only 20 min. Despite the very short practice time, we attributed some results to later learning phases, which we justified with the very simple motor task (sequence of only four elements), the errors made (error rate decreased in the course of the learning time), the gaze behavior (visual control became unimportant), and the performance (performance approached its asymptote). Nevertheless, we must admit that this simplification is a limitation of the study. Although the task had consistent stimulus–response mappings (i.e., the same motor sequence consisting of four elements was repeatedly played with the same fingers), we expected positive performance–ability associations, which *increased* over the course of learning. This is because the task was designed to continuously demand maximum effort from the participants. As a result, movement speed becomes more and more important, which is strongly related to cognito-motor functions, as suggested by studies on age-related slowing [55,56] We also noted that other forms of motor sequence learning require cognito-motor skills in a similar way and that this is not a unique feature of music making. Therefore, we hypothesized that comparable associations would occur in a second, music-independent motor sequence learning task.

The results were hypothesized to be useful in deriving methodological and content-related implications for interventions to promote healthy aging. Furthermore, the results may also reconcile conflicting findings about the efficacy of particular cognito-motor interventions because observed effects may be phase- and/or time-dependent.

## 2. Materials and Methods

### 2.1. Participants

Eighty-six older adults (mean age = 72.5, SD = 3.5; 41 men, 45 women) participated in this experiment. All subjects were healthy, without any signs of neurological, psychological, or motor problems. Most of them had not played a musical instrument in their lives, while N = 37 had some piano experience (median = 48 months; range = 12–180 months) with a cumulative practice time of 1061.3 h on average (SD = 676.2 h; range = 120–2944.3 h). Participants with extensive piano experience (cumulative practicing time > 4000 h) were excluded from the study. The General Musical Sophistication measured with the Gold-MSI in our sample averaged 56.8 (SD = 17.1) and was therefore lower than in the general population (mean = 81.6, SD = 20.6; [57]).

### 2.2. Procedure

Within one session, first, a piano-related and, later, a piano-unrelated motor sequence should be learned. Both tasks consisted of 20 trials of 20 s with a 30 s break between trials (Figure 1). Each motor sequence consisted of four elements, which had to be executed alternately with the right and left hand. A comprehensive test battery, including cognitive, musical, and motor assessments, was applied (see below). In order to keep mental fatigue to a minimum, the test battery was spread over the entire experiment in a fixed order. The whole experiment took approximately 100 min.

### 2.3. Motor Sequence Task

Both motor sequence tasks were implemented in *PsychoPy* [58], an open-source software for creating experiments. Participants saw the corresponding sequence graphically on display and pressed the highlighted keys from top to bottom *as quickly and accurately as possible* on the piano or response pad. When participants reached the bottom of the display, they automatically started at the top again, as if in a loop. Before the test, a training trial with a different sequence order was completed. The piano sequence performance was recorded using a Yamaha MIDI piano (Figure 2A). In this test, the piano keys were colored to allow better orientation on the keyboard. Each finger was assigned a key. The keys had to be pressed in a fixed order: right ring finger, left middle finger, right thumb, and left thumb in a repeated order. The piano-unrelated sequence was performed using a response pad (Cedrus RB-844, Cedrus Corporation, San Pedro, CA 90734 - USA; Figure 2B). In this task, the keys had to be pressed alternately with the right and left index fingers.

The number of correct key presses per trial was defined as the dependent “performance” variable. The total amount of errors and the error rate (errors divided by total key presses) were computed, with errors also classified as *wrong key* (pressing a key which is not part of the sequence) or *wrong order* (pressing a key which is part of the sequence but not in the correct order). 

### 2.4. Gaze Behavior

Participants’ gaze was captured with a video camera. Gaze changes (i.e., from the display to the keyboard/response pad or vice versa) were counted for each trial retrospectively. Due to technical problems, the gaze behavior of twelve participants could not be analyzed.

### 2.5. Test-Battery

Clicking speed

The participants were asked to alternately press two keys a few centimeters apart as quickly as possible with the right (Click_R), left (Click_L), and both index fingers (Click_B). Each condition was performed twice, with the better score used for the analysis.

Purdue Pegboard

The Purdue Pegboard test [59] consists of two unimanual and two bimanual conditions. First, participants were asked to place as many pegs as possible in a vertical row within 30 s using the right (PP_R), left (PP_L), or both hands simultaneously (PP_B). Finally, the participants had to assemble as many “towers” as possible within 60 s (PP_A), which consisted of a pin, a washer, a collar, and another washer. Each condition was performed twice. The better score was used for the analysis. 

Coding

Coding (C), part of the Wechsler Adult Intelligence Scale-IV (WAIS-IV; 60), pairs numbers from 1 to 9 with a unique symbol. Within 120 s, the participants had to draw as many symbols as possible under the corresponding number.

Symbol Search

Symbol Search (SS), also part of the WAIS-IV [60], required participants to search through a row of five symbols to determine whether one of two target symbols occurs or not. The score is the number of correct rows within 120 s.

Number Connection Test

A German version of the Number Connection Test (“Zahlen-Verbindungs-Test”; ZVT) from Oswald [61] was used. In this test, the subjects had to connect numbers in ascending order with a pencil as quickly as possible. The score corresponded to the time it took to connect all the numbers.

Reading letters

In this test [62], participants had to read four different sequences of letters as quickly as possible. The shortest time in four attempts corresponded to the participant’s score.

Trail-Making-Test

A Trail-Making-Test (based on [63]) was also used. In the first condition, participants had to connect randomly arranged numbers with a pencil as quickly as possible in ascending order (TMT_A). In the second condition (TMT_B), numbers and letters were to be connected alternately but in ascending order (i.e., 1-A-2-B-3-C, etc.). The score corresponded to the time it took to connect all the numbers (and letters).

Digit Span

The participants had to repeat increasingly longer sequences of numbers in the same (i.e., forward; DSP_F), reversed (DSP_R), or ascending (i.e., sequential; DSP_S) order. Two different number sequences were presented for each length. The task was aborted if both sequences of the same length were repeated incorrectly. The score was equal to the total number of correctly repeated sequences.

Corsi Blocks

Based on Milner [64], a computerized version of the Corsi Block test [65] was used. The participants saw randomly arranged rectangles on a screen. They had to select increasingly longer sequences of flashing rectangles in the same (BS_F) or reversed (BS_R) order. Two different rectangle sequences were presented for sequence length. The task was aborted if both sequences of the same length were repeated incorrectly. The score was equal to the total number of correctly repeated sequences. Due to technical reasons Corsi Blocks could not be applied for 22 participants.

Gold-MSI Questionnaire

The Goldsmiths Musical Sophistication Index (Gold-MSI; [57]) is a self-report inventory that measures various facets of musical sophistication, including active engagement, perceptual skills, musical training, singing skills, emotional response to music, and overall musical sophistication.

Piano experience questionnaire

The participants were asked how long and how often they had played the piano in different periods of their lives. The total duration was used for the analysis.

### 2.6. Statistics

All statistical analyses were performed using the software *R* [66]. Latent factors representing processing speed, psychomotor speed, working memory and dexterity were extracted from the results of the neuropsychological test battery. As the underlying construct of each test was known the factor extraction was carried out by means of confirmatory factor analysis using the package *lavaan* [67]. Coefficients were estimated using Maximum Likelihood estimation. 

For further analysis, we implemented Bayesian multilevel models using the R package *brms* [68]. In the first step, based on the participants’ performance data, individual learning curves were reconstructed. As performance usually shows a gradual decrease in improvement, exponential models were specified. 

By this, α (i.e., baseline performance), β (i.e., asymptote), and γ (i.e., learning rate) were estimated. All parameters were allowed to vary across participants and to correlate with each other (indicated by the nesting structure). The modeled performance data was then standardized for each trial (1 unit = 1 SD) to allow for comparability of beta coefficients across learning. Finally, standardized sequence performance was predicted by a latent cognito-motor factor, musical sophistication, and piano experience. Each predictor was tested in a separate model, controlling for individual learning rates. In order to be able to analyze each trial separately, trials were treated as factors. 

For both tasks, errors and gaze behavior were analyzed. While the former was also analyzed with exponential models, for the latter we applied piecewise nonlinear modeling with individual change points. The change point was a model parameter to be estimated and allowed to vary across participants. It represents the trial number at which the number of gaze changes approached zero, i.e. when visual control becomes negligible for the respective subject when performing the task. The Markov Chain Monte Carlo (MCMC) estimation was performed with four chains. Each model was run with 5000 iterations, including a warm-up phase of 2000 iterations. All effects are reported with a 95% credible interval.

## 3. Results 

### 3.1. Latent Factor Analysis

Only the model with the best fit is explained below and shown in Figure 3. Reading letters, BS_F as well as TMT_B were discarded due to inconclusive loadings. DSP_S were allowed to load on both working memory and processing speed latent factors. The confirmatory factor analysis showed good model fit (CFI = 0.941, TLI = 0.926, IFI = 0.943, RMSEA = 0.072, SRMR = 0.070; see Lüdecke et al. [69] for recommended fit indices). 

Processing speed, dexterity, and psychomotor speed were significantly correlated with each other (all *p* < 0.001; see Figure 3). Working memory was only correlated with processing speed (r = 0.44, *p* = 0.004).

### 3.2. Errors

The total number of errors in the piano task increased slightly, with an average of 1.73 (±3) errors on the first trial to 4.22 (±10.97) on trial 20 (Figure 4). The error rate decreased from an average of 15.04% (±23.52) to 6.68% (±13.41). In the response pad task, the total errors fell very slightly from an average of 2.73 (±6.29) in trial 1 to 1.51 (±3.08) in trial 20. As with the piano task, the error rate fell from 10.5 (±18.22) to 2.35 (±6.79) during the learning phase. 

The rate at which errors in the piano task decreased was positively correlated with dexterity (r = 0.31, *p* = 0.003). The rate at which errors decreased in the response pad task was also positively correlated with dexterity (r = 0.29, *p* = 0.006).

### 3.3. Gaze Behavior

The number of gaze changes, as well as gaze changes per key press, decreased across trials (Figure 5). Piecewise nonlinear modeling with individual change points revealed largely negligible visual control at around trial 7.06 [5.69, 8.16] and 4.92 [4.43, 5.42] for the piano-related and music-unrelated sequence task, respectively.

The rate of the gaze behavior (i.e., the number of trials in which the change point was reached) in the piano task correlated with dexterity (r = 0.23, *p* = 0.054) and working memory (r = 0.23, *p* = 0.056).

The rate of gaze behavior in the response pad task showed a positive trend to correlate with psychomotor speed (r = 0.21, *p* < 0.075).

In summary, gaze rate correlated only weakly with some cognito-motor variables. Better abilities enabled the participants to solve the task earlier without visual control.

### 3.4. Motor Sequence Performance 

The raw performance data of both tasks showed characteristics of exponential functions (Figure 6 left). Models were compared by calculating their differences in expected log pointwise predictive densities (elpd-differences). High elpd-values indicate high predictive performance. For the music-independent sequence task, exponential modeling yielded a better fit than linear (−624.3), quadratic (−270.0), and cubic modeling (−70.5). In the piano sequence task, exponential modeling resulted in a better fit than linear (−499.4) and quadratic (−109.5) models but a worse fit than cubic modeling (+74.4). Due to the higher number of parameters specified in cubic models as well as a theoretical justification (see [70]), we selected the exponential model for the analysis.

Graphical posterior predictive checks indicated good model fit. The convergence of the model was determined using Rhat values. If Rhat is significantly greater than 1 (a value of 1.1 is often assumed as a cut-off), it is assumed that the chains did not converge. However, the present algorithms were satisfactory with Rhat values of ≤ 1.01. In addition, the models yielded visually well-mixed chains, and no divergent transitions occurred.

In comparison to the response pad task, the piano task seemed more difficult and showed a lower general learning rate (γ_Piano_ = −1.98 [−2.22, −1.75] vs. γ_RespPad_ = −1.10 [−1.29, −0.93]). According to Pearson correlation coefficients, individual learning rates (r = 0.37, *p* < 0.001), baseline performances (r = 0.44, *p* < 0.001), and asymptotes (r = 0.69, *p* < 0.001) for both tasks were positively associated. 

The learning rate of the piano task was positively correlated to dexterity (r = 0.22, *p* < 0.001), psychomotor speed (r = 0.30, *p* < 0.001), working memory (r = 0.09, *p* < 0.001), and processing speed (r = 0.27; *p* < 0.001). 

The learning rate of the response pad task was positively correlated to dexterity (r = 0.24, *p* < 0.001), psychomotor speed (r = 0.41, *p* < 0.001), working memory (r = 0.26, *p* < 0.001), and processing speed (r = 0.40; *p* < 0.001).

In summary, the learning rate correlated significantly, but only weakly to moderately, with the cognito-motor variables, thus allowing simultaneous inclusion of ability and learning rate in the regression models without causing multicollinearity problems.

### 3.5. Performance-Cognito-Motor Relationship

Processing speed, psychomotor speed, and dexterity were positively associated with music-related and music-unrelated motor sequence performance. The strength of the associations changed over the learning phase, with relatively strong associations in trial 1, the weakest association in trial 2, and a subsequent increase in associations approaching a plateau (Figure 7).

Higher working memory capacity predicted better piano-related performance in trial 1 (0.18 [0.03, 0.32]), decreased to 0.15 [−0.07, 0.36] in trial 2, and then steadily increased again, reaching a standardized effect size of 0.22 [0.00, 0.43] in trial 20. Similar patterns were found for processing speed (starting at 0.30 [0.16, 0.44]), dexterity (0.29 [0.16, 0.44]), and psychomotor speed (0.34 [0.21, 0.49]), yielding, respectively, an effect size of 0.37 [0.16, 0.58], 0.38 [0.17, 0.59] and 0.49 [0.30, 0.69] at trial 20. The course of the associations between piano playing and cognitive abilities could only be partially measured, as the associations appear to strengthen further with prolonged practice beyond trial 20.

On the other hand, the time-variant associations between response pad performance and cognito-motor skills, which plateaued around trial 15, were probably fully captured. Better processing speed (0.24 [0.10, 0.39]), dexterity (0.19 [0.05, 0.57]), and psychomotor speed (0.28 [0.14, 0.42]) predicted higher initial response pad performance. After a short decrease at trial 2, the coefficients stabilized at 0.37 [0.16, 0.57], 0.35 [0.15, 0.56], and 0.52 [0.33, 0.71], respectively. Working memory capacity could not meaningfully predict response pad performance in any trial. 

### 3.6. Performance-Musicality Relationship

Both tasks did not show any relationship to any music variable.

## 4. Conclusions

Working memory, processing speed, psychomotor speed, and dexterity were positively associated with piano performance. The associations between performance and ability allow three important conclusions to be drawn: First, under certain circumstances, performing musical pieces can be regarded as a consistent high-performance task. Although stimulus–response mappings are invariant, the best possible performance is unlikely to be achieved by automatic control processes. Our results suggest the exact opposite: in the course of learning, the importance of cognitive processes for performing actually increases. This refutes traditional automaticity frameworks [8,22] and confirms more recent findings from movement sciences [26,27]. Since the learning time was very short, an attempt should be made to refute this conclusion in a study with a longer learning time. Second, practicing novel skills requires cognito-motor abilities to different degrees. And third, the relationship between performance and cognito-motor abilities changes over the practice period. Independent of ability, we found relatively strong associations in trial 1, the weakest association in trial 2, and a subsequent increase in associations.

Although both tasks are characterized by fixed stimulus–response mappings that were consistently practiced, an underlying three-stage learning process cannot be assumed, given the findings on performance, gaze–behavior, and performance–ability correlations. Instead, the results suggest that performing consistent high-performance tasks (i.e., executing sequences “as fast and accurately as possible”) is linked to increasing cognitive and motoric demands that approach a plateau. An exception might be sight-reading (i.e., the first attempt at the piano sequence), which, despite the early learning phase, shows relatively high demands on working memory (0.18 [0.03, 0.32]), processing speed (0.30 [0.16, 0.44]), dexterity (0.29 [0.16, 0.44]), and psychomotor speed (0.34 [0.21, 0.49]. Interestingly, Kopiez and Lee [42] found similar strong associations (see introduction), despite including piano majors, graduates, and postgraduates from a music university. Analogous patterns with relatively high initial cognito-motor demands were also found in the non-music-specific task. This indicates that the initial learning phase, in particular, in which the individual familiarizes themselves with the task and selects a goal and an action plan [2,17], requires high cognito-motor effort. In comparison to the piano task, however, the working memory demand in the response pad task was not statistically meaningful. One possible explanation is that due to the higher degrees of freedom (DoF), the piano task involved greater complexity in goal and action selection than the task with the two-finger reaction pad (10 vs. 8 DoF). In addition, piano tasks may be effector-dependent and require egocentric representations, meaning that the movement is computed with respect to a body-related reference frame [71]. Therefore, sight-reading must first be translated into internal coordinates (i.e., selecting the corresponding finger) before a movement can be initiated. Possibly, the response pad task requires an extrinsic, spatial coordinate frame (allocentric) that may be less complex to process and can be performed simply by reacting to the displayed notes (see *Reaction mode* in 15). The subsequent decreasing associations between performance and ability in the second trial can be explained by efficient recycling of established mental representations; the subsequent increase could be due to the higher demands of processing kinesthetic and auditory feedback [17] as well as the higher rate of initiated movements and concatenated motor chunks [72].

The gaze behavior results indicate a decrease in visual control over the motor sequence learning period. After 5–7 trials, most participants no longer needed visual information to perform the required movement sequence. We interpret this finding as a significant influence of episodic memory function only during the early learning phase, in which explicit knowledge, such as the sequence order, must be acquired [29]. In the course of the learning process, movement sequences are increasingly consolidated in procedural memory, and explicit information loses importance [10]. After a certain time, the participants are often no longer capable of explicitly recalling the motor sequence [73,74]. Strictly speaking, however, the acquisition of explicit knowledge is not a necessary prerequisite for solving or learning motor tasks. Implicit learning may be sufficient to master a motor task without participants being able to describe the facts and rules of the task. However, the strength of the interaction and the dependence of explicit and implicit learning are beyond the scope of this article and have been discussed elsewhere [2,75,76].

Neither task performance was associated with music experience or sophistication. It is surprising that the piano experience (i.e., the amount of lifetime practice) was not even transferable to the piano task of the experiment. Perhaps the piano condition was too dissimilar to a real piano situation, e.g., due to the unusual notation we used. However, previous studies have shown that experienced pianists have an above-average ability to learn implicit sequences, even beyond their area of expertise [74,77]. Thus, it is possible that the range of piano experience in our sample was too narrow to be linked to positive transfer effects on general motor sequence learning.

### 4.1. Implications for Interventions Promoting Healthy Aging

Since transfer effects are more likely the greater the overlap between the trained and transferred skill [53], the positive associations between performance and abilities found here suggest that learning to play the piano could induce positive cognitive and motoric effects in older adults. This assumption is supported by findings of long-term music interventions showing that music can decelerate age-related cognitive decline and thus support healthy aging [33,38,39,78,79]. With the second task, we were able to show that similar effects can be expected when learning other non-musical motor skills, as similar cognito-motor demands are made. Accordingly, making music is just one of many ways of slowing age-related decline. This is also even more generally supported by the results of a recent fMRI study showing that a frontoparietal network consisting of bilateral inferior frontal gyrus and bilateral intraparietal sulcus supports skill learning [80].

Based on the strength of the associations between piano performance and cognito-motor abilities, learning to play the piano (i.e., acquiring music repertoire) would be expected to have small effects on working memory (standardized β-coefficient below 0.25) and small-to-medium effects on processing speed, dexterity and psychomotor speed with standardized β-coefficient up to 0.5 SD. We were able to demonstrate relatively similar effect sizes in an earlier randomized controlled trial with 156 older adults [33]. After 12 months of piano lessons, the participants improved little in working memory (0.24 [0.04, 0.45]) and moderately in processing speed (0.33 [0.18, 0.49]) and dexterity (depending on the Purdue Pegboard condition, the standardized β-coefficients ranged from 0.25 [0.05, 0.46] to 0.49 [0.29, 0.69]).

Two implications for music therapy interventions can be derived from the dynamics of the relationship between performance and cognitive abilities. On one hand, longer interventions can have a greater impact. This is not only because more time is available to adjust to the requirements but also because the dose–response relationship seems to follow an exponential function in which the full transfer potential can only be exploited after some time [81]. A previous meta-analysis specifically addressing dose–response relationships underlined this conclusion and revealed that music therapeutic effects increase with the number of sessions provided: small effects were seen after 3 to 10 sessions, medium effects after 10 to 24 sessions, and large effects after 16 to 51 sessions [82].

On the other hand, the intervention and the training content should not be consistent, e.g., by repeatedly rehearsing a piece at a comfortable tempo. Granted, such consistent content could have other (positive) effects, such as improving musical aspects (e.g., rhythm and articulation) or triggering emotion regulation processes. However, if the intention is to promote cognito-motor functions, the training protocol should be adaptive and/or include consistent high-performance tasks. For example, learning new, sufficiently complex pieces of music or playing at a constantly challenging tempo makes high cognito-motor demands, as the results presented here show. However, whether this training approach actually achieves positive effects in older people and can contribute to healthy aging must first be investigated in long-term studies.

### 4.2. Strengths and Limitations

Previous studies [7,29,31,34,41,65,77,83,84] analyzed performance–ability correlations *over time*, for example, by correlating working memory with raw or smoothed performance data at each measurement *time point*. Accordingly, individual learning rates are neglected and the contribution of certain cognitive functions cannot be necessarily transferred to a learning *stage*. We overcame this issue by statistically controlling for individuals’ learning rates.

Researchers often confuse task-specific effects with task-general effects [85]. In other words, cognitive test results are equated to cognitive abilities. However, this is misleading and will lead to unreliable results. In the present study, we analyzed abilities derived from the Cattell–Horn–Carroll (CHC) theory of intelligence [86] in predicting motor sequence performance. The CHC framework is a highly established theory that provides important information on how human cognitive abilities are structured. Furthermore, it proposes and defines a number of broad abilities (for example, “Psychomotor speed”) and identifies associated narrow abilities (for example, “Speed of limb movement”). In the present study we considered abilities as latent constructs, i.e., they are not directly observable and must be derived from other measurable variables. To this end, we conducted a confirmatory factor analysis with at least three indicators loading to one factor.

We aimed to investigate the associations between performance and ability throughout the entire learning process. Although our participants approximated their asymptotic performance, learning complex motor skills, such as mastering a musical instrument, can take decades and thousands of hours of practice [87]. In addition, research has shown that sleep can facilitate the consolidation of motor memory, with small to moderate effect sizes depending on the motor task, e.g., g = 0.48 for finger-tapping tasks [88]. Therefore, our results are more representative of early learning and should only be cautiously generalized to the late learning periods.

And finally, despite the fact that we found meaningful associations between motor sequence performance and cognito-motor skills, we can only point to an existing *potential* of both motor tasks to induce transfer effects. Longitudial studies are required to infer causality and to verify the positive cognitive and motor effects.

## Figures and Tables

**Figure 1 brainsci-14-00405-f001:**
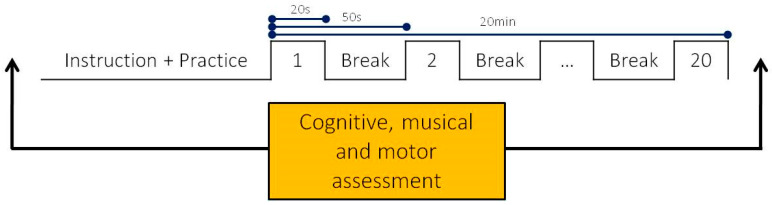
Structure of the motor sequence tasks.

**Figure 2 brainsci-14-00405-f002:**
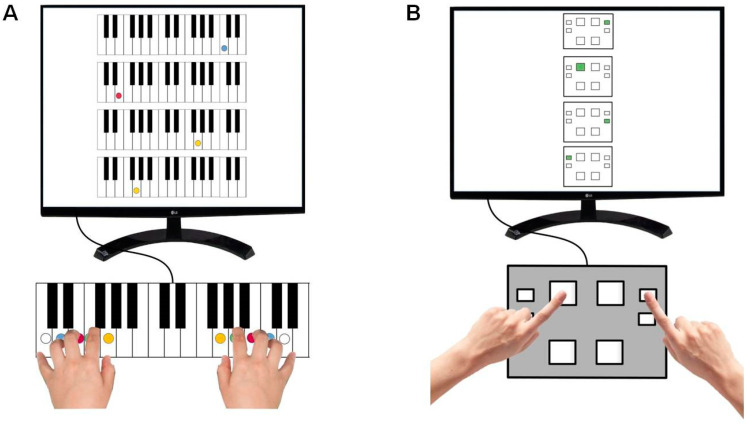
Piano-related (**A**) and piano-unrelated (**B**) motor sequence. The highlighted keys must be pressed alternately with the right and left hand as fast and accurately as possible.

**Figure 3 brainsci-14-00405-f003:**
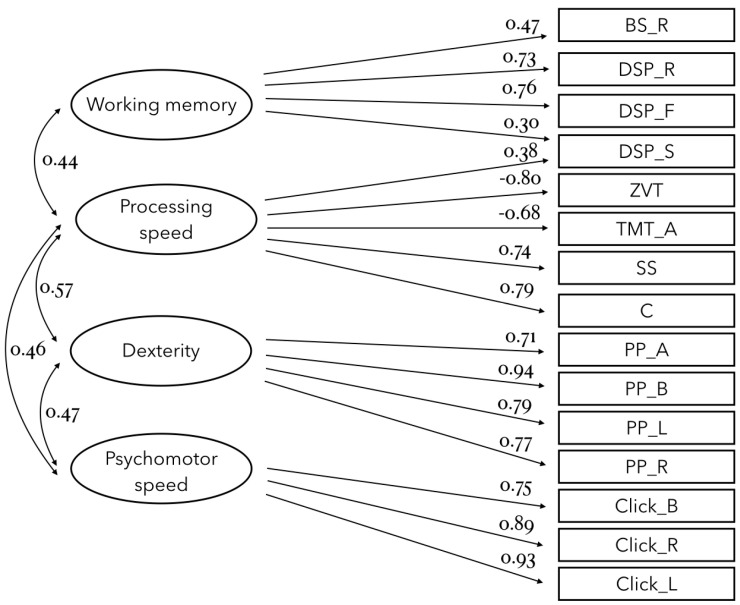
Results of the confirmatory factor analysis. Abbreviations: BS_R, Corsi Block Span reversed; DSP_R, Digit Span reversed; DSP_F, Digit Span forward; DSP_S, Digit Span sequential; ZVT, Number Connection Test; TMT_A, Trail-Making-Test ascending; SS, Symbol Search; C, Coding; PP_A, Purdue Pegboard assembly; PP_B, Purdue Pegboard both hands; PP_L, Purdue Pegboard left hand; PP_R, Purdue Pegboard right hand; Click_B, Clicking speed alternating hands; Click_R, Clicking speed right hand; Click_L, Clicking speed left hand.

**Figure 4 brainsci-14-00405-f004:**
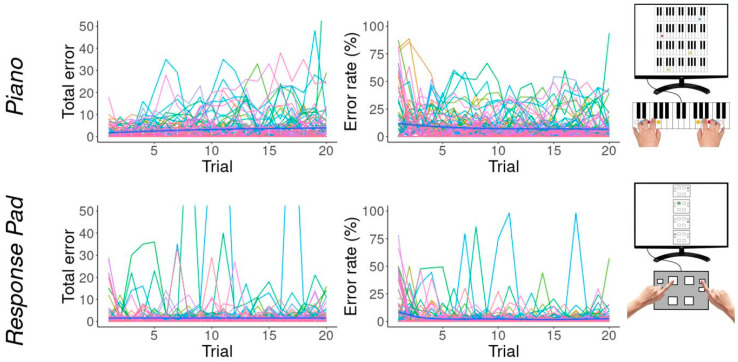
Total number of errors (**left**) and error rate (**right**) per trial. Each color represents one participant.

**Figure 5 brainsci-14-00405-f005:**
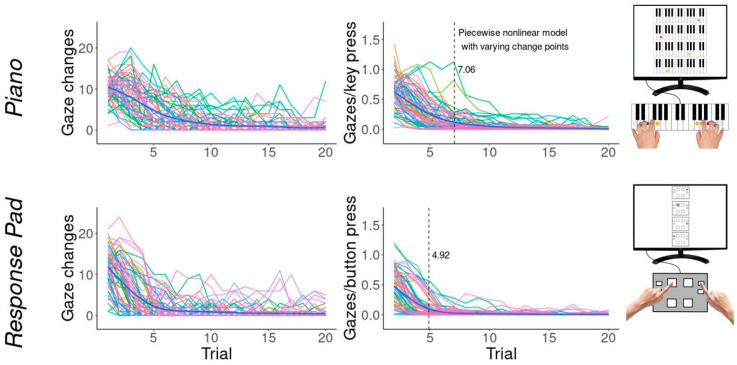
Absolute (**left**) and relative (**right**) number of gaze changes per trial. Each color represents one participant.

**Figure 6 brainsci-14-00405-f006:**
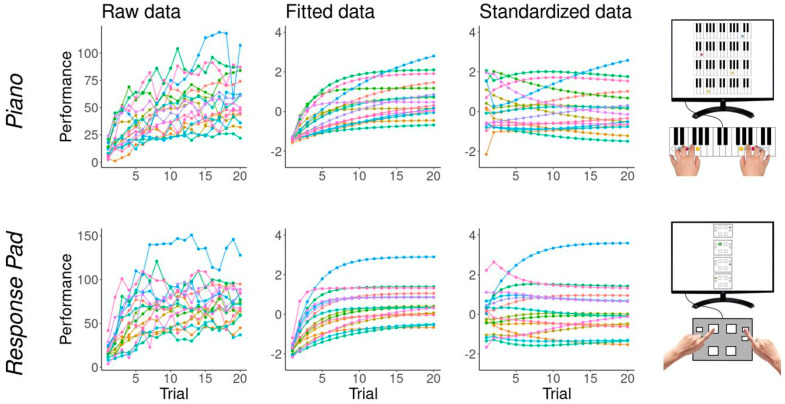
Performance of both sequence tasks, shown as raw data (**left**), as data fitted with exponential functions (**middle**) and as additionally normalized data for each trial separately (**right**). Performance of 17 randomly selected participants is shown. Each color represents one participant.

**Figure 7 brainsci-14-00405-f007:**
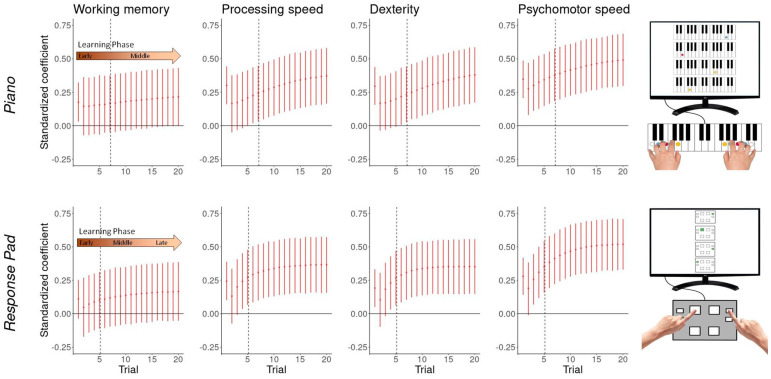
Performance-ability associations change over the learning period.

## Data Availability

The original data presented in the study are openly available at https://osf.io/x3zuf/.

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
