# Peer review of "What Does It Take to Play the Piano? Cognito-Motor Functions Underlying Motor Learning in Older Adults"

_brainsci, 2024, doi:10.3390/brainsci14040405_

Round 1
Reviewer 1 Report
Comments and Suggestions for Authors
The authors tested the associations between cognomotor abilities and performance on a piano-related and a non-musical sequencing task and found a pattern of changing ability-performance relationships over the course of learning the two tasks. They argue that their results can inform the development of musical/motor interventions to encourage healthy ageing via transfer of learning effects.
This is an interesting study and it has some notable strengths. These include the use of a comprehensive battery of cognitive and motor tasks to assess cognomotor abilities, the extraction of latent ability factors based on the test battery to use a predictors, a tractable task, detailed characterizations of participants’ performance and rates of learning, and appropriate use of statistical models to describe ability-performance relationships during practice. However, the study also has some notable weaknesses. Chief among these are the simplistic nature of the task, the limited amount of practice given to participants, and the major disconnect between the interpretation of the findings, which focus on the transfer of learning to improve cognitive and motor function, and the objective of the study, which attempted to simply describe how ability-performance relationships changed over the course of a short bout of practice of two relatively simple tasks. I detail some of my concerns about the study’s weaknesses below and I raise questions I would like to see the authors answer.
Modifying the task so that participants’ performance could come close to plateau during a single practice session clearly made the experimental study more tractable, but at the expense of confining the study’s findings to similarly difficult tasks. Research shows that task difficulty interacts with a number of variables that influence motor skill learning (see Guadagnoli & Lee’s 2004 challenge point framework and recent iterations of it, for example). Moreover, even though the participants’ performance generally approached plateau, the study’s findings also pertain only to the earliest stage of learning. The authors need to articulate more clearly why they think their findings can provide insight into what happens at later stages of learning, even though they describe early, middle, and late phases of practice in their description of the results. After all, participants only performed 20 20-second trials for a total practice time of just under 7 minutes. Furthermore, by simplifying the tasks so that participants could reach plateau in a single short practice session, the authors minimized the cognitive and motor resources participants needed to deploy to succeed on the tasks. Consequently, I do not think the study’s findings will generalize beyond the current tasks.
Guadagnoli, M. A., & Lee, T. D. (2004). Challenge point: A framework for conceptualizing the effects of various practice conditions in motor learning. Journal of Motor Behavior, 36(2), 212–224.
The authors also need to articulate more clearly why they expected the performance-ability associations to increase over time given the tasks’ consistent stimulus-response mappings. If this paradigm generalizes to more difficult tasks and to later stages of learning, and following Ackerman’s logic, we would expect to see working memory associations with practice decrease over practice, processing speed associations increase and then decrease, and dexterity and psychomotor speed associations to increase over practice even though participants invested maximum effort throughout their practice.
Why did the authors only use the number of correct key presses per trial as the dependent variable? Didn’t the errors covary with the number of correct key presses? I do not understand why the error rate was so low (<0.5%) if participants improved across practice trials. Further, why not use movement time as an additional dependent variable? For example, the authors note that better cognomotor skills enabled participants to solve the task faster without visual control (line 326-27).
The associations between the cognomotor abilities and performance on trial 1 seems anomalous. Why do the associations drop from trial 1 to trial 2 so sharply and then increase again? In many cases, the magnitude of the ability-performance relations on trial 1 are very close to the relations on trial 20, particularly for the piano-related task. This pattern of finding strikes me as very odd and requires explanation.
Why did the authors administer the neuropsychological and psychomotor tests before and after practice? Where are the post-practice data?
The conclusion section seems to relate to a different paper with a different set of findings from the ones presented here. The authors did not test for transfer effects and yet their conclusion focuses largely on evidence for such effects. There is a clear disconnect between the interpretation of the findings and the findings themselves. Consequently, the conclusion section needs extensive rewriting.
The authors write in the second sentence of the conclusion, lines 384-388 “As transfer necessitates sufficient overlap between the trained and the transfer skill [47], learning the piano shows potential to induce beneficial cognitive and motoric effects in older adults. This outcome corroborates findings of long-term music interventions showing that music can decelerate age-related cognitive decline and thus supports healthy aging [27,37,38,63,64].” How did they reach this conclusion? They did not test this.
On lines 395-405, the author also write “The associations between performance and ability allow three important conclusions to be drawn: First, reciting musical pieces can be regarded as a consistent high performance task. Although stimulus-response mappings are invariant, the best possible performance cannot be achieved by automatic control processes but remains cognitively demanding. This refutes traditional automaticity frameworks [7,17] and confirms more recent findings from movement sciences[21,22].” I do not understand how the authors reached this conclusion. Participants received less than 7 minutes of practice and were never going to automate their performance after such limited practice. Moreover, skilled musicians frequently describe playing well-known pieces of music automatically and many have reported experiencing flow states in which they feel someone else or something else is controlling their movements.
The authors go on to write “Second, practicing novel skills requires cognomotor abilities to different degrees. Based on the strength of the associations between piano performance and cognomotor abilities, learning to play the piano (i.e. acquiring music repertoire) would be expected to have small effects on Working memory (standardized β-coefficient below 0.25), and small-to-medium effects on Processing speed, Dexterity and Psychomotor speed with standardized β-coefficient up to 0.5 SD.” Again, I do not understand why the authors focus on implications the study did not test. Moreover, if we stay true to the traditional distinction between abilities and skills described by Fleishman (e.g., 1966, 1972) and endorsed by Ackerman, we would expect very little transfer from a practiced skill to the abilities that support that skill. Fleishman and Ackerman both believed that abilities represent stable traits that are resistant to change. The authors need to describe how their conception of abilities leads to the expectation that abilities are mutable.
Fleishman, E. A. (1966). Human abilities and the acquisition of skill: Comments on professor Jones’ paper. In E. A. Bilodeau (Ed.), Acquisition of skill (pp. 147–167). New York: Academic Press.
Fleishman, E. A. (1972). On the relation between abilities, learning, and human performance. American Psychologist, 27(11), 1017–1032.
Based their findings of increasing ability-performance relations over practice, the authors go on to conclude that longer music interventions can have greater impact and that intervention and training content should not be consistent. They suggest that if the intervention is to promote cognomotor functions, the training protocol should be adaptive and/or include consistent high-performance tasks. Again, I do not know how they reached these conclusions based on the data they present.
The authors also write on lines 430-432 “For example, learning new, sufficiently complex musical pieces or playing at a consistently challenging tempo can be very effective in achieving positive cognomotor effects, as the results presented here show.” Which results show this effectiveness?
In discussing the gaze data on lines 437-440, the authors note that “In the course of the learning process, movement sequences are increasingly consolidated in procedural memory and explicit information loses importance [9]. After a certain time, the participants are often no longer able to recall the motor sequence [68,69].” This runs counter to their earlier argument that their task cannot be automated with practice.
On lines 454-459, the authors write “Although both tasks are characterized by fixed stimulus-response mappings that were consistently practiced, an underlying three-stage learning process cannot be assumed given the findings on performance, gaze-behavior, and performance-ability correlations. Instead, the results suggest that learning consistent high-performance tasks (i.e. performing sequences “as fast and accurate as possible”) appears in a continuum with increasing cognitive and motoric demand that is approaching a plateau.” This conclusion only applies to the simple task used in the current experiment and only to the earliest phase of skill acquisition.
At the end of the conclusion section, the authors highlight the disconnect between the objectives of the study and their interpretation of their findings:
Lines 485-487 “In the present study we analyzed abilities derived from the Cattell–Horn–Carroll (CHC) theory of intelligence [76] in predicting motor sequence performance.”
Lines 497-498 “Therefore, our results are more representative of early learning and should only be cautiously generalized to the late learning periods.”
“And finally, despite the fact that we found meaningful associations between motor sequence performance and cognomotor skills, we can only point to an existing potential of both motor tasks to induce transfer effects. To infer causality and to verify the positive cognitive and motor effects longitudinal studies are required.”
Comments on the Quality of English LanguageI found the quality of writing fine. The authors just need to correct a couple of awkward word choices and phrasings.
Reviewer 2 Report
Comments and Suggestions for Authors
This is an interesting paper. The paper is well written, though the readability and understandability could be improved by better explaining some technical terms. The methodology seems also to be sound, but should be explained also somewhat more intuitively. One major strength of the paper is that it tries to reconcile previous found conflicting results. Overall the paper meets high academic standards, but should be rewritten somewhat to improve the readability.
General remarks
· The contents are very interesting and quite relevant. This holds in particular for the description of the cognitive process theory and the performance-ability associations.
· The style of writing is quite mature and rather fluent.
· The referencing style is at the needed standards.
· Some technical terms must be explained much more in detail at first occurrence in the text. Examples are: discrete sequence production task, tower of Toronto, etc. (see below also in detailed comments)
· The paper provides a strong theoretical and empirical background with a strong positioning of the research within the broader research field.
· The specific aim of the study is not totally clear. This should be given more explicitly.
· The statistical methods must be explained somewhat more in intuitive terms so as to improve the readability and understandability of the paper. The technicity of the methods is quite sophisticated but it leaves the untrained reader with a feeling of lack of certainty at some moments. There is also the danger of (unintentional) pedantry by introducing terms or methods that are explained in the slightest way.
Detailed comments
· Line 35: suggestion for additional reference, very related to the common topics of this paper: Reybrouck, M. & Schiavio, A. (2024) Music performance as knowledge acquisition: a review and preliminary conceptual framework. Front. Psychol. 15:1331806. doi: 10.3389/fpsyg.2024.1331806
· Line 74: provide a short intuitive description of what is meant exactly with “discrete sequence production task” at first appearance of the term.
· Line 81: explain very shortly what is meant with the Tower of Toronto. Even is this may seem to be a very well-known test, it is not common knowledge for all readers who are not psychologists.
· Line 149: even though the aims of the study are well described, the specific research questions should be stated more explicitly.
· Line 188: explain very shortly what PsychoPy is.
· Line 197: what is a Cedrus respose pad? Explain shortly.
· Lines 213: the abundance of numerical numbering is very heavy qua layout. Perhaps bulleting is a better option
· Line 271: there are several methodologies for factorial analysis (Factor analysis, Principal Component Analysis, etc.). Explain very shortly why confirmatory factor analysis is chosen, and how it is to be distinguished from the other methodologies. The used formulas must be explained more clearly.
· Line 273: what does brms stand for? Introducing abbreviations without explaining them may be seen as (unintended) pedantry.
· Line 284: why not introducing Figure 5 at an earlier place and give it the name Figure 1. It helps the reader when the figures are numbered in order of occurrence.
· Lines 288 ff: this is rather difficult to understand. Explain more clearly how to understand the individual change points.
· Figure 3: Please provide the explanations of the abbreviations in the figure caption. There is too much needed searching efforts by the reader now.
· Line 330: what are elpd-differences. Please explain a little.
· Lines 337: the Bayesian approach is quite difficult to understand. What are Rhat -values? Please explain somewhat more intuitively. This whole paragraph is very hard to understand.
· Line 407: what does RCT mean? Please explain.
· Lines 472 ff: please explain somewhat more the egocentric representations. Very difficult to understand what is meant here.
Line 486: explain shortly what is meant with the CHC theory of intelligence.
Round 2
Reviewer 1 Report
Comments and Suggestions for Authors
I thank the reviewers for their responsiveness to my original concerns and their thoughtful responses. Although I do not agree with everything they have said, they have made strong cases for their positions. I do not have any concerns with the revised manuscript and think it warrants publication. However, I would urge the authors to reconsider their first conclusion in the conclusion section - that this type of consistent high performance task cannot be achieved by automatic control processes. I think the only way to make that conclusion with any surety is to run this study with much more practice. The performance itself might not change or might not change very much (i.e., in terms of number of correct key presses, errors) but that does not mean the processes underlying performance do not change at later phases of practice. It's quite possible that performance stays the same but the abilities that support performance get reorganized. I do not make implementing this suggestion a condition of my recommendation to accept the paper. I simply put it out there for the authors to consider further.
Author Response
- Dear reviewer,
we have made the conclusion more vague and adapted it:
“Although stimulus-response mappings are invariant, the best possible performance is unlikely to be achieved by automatic control processes” […] “Since the learning time was very short, an attempt should be made to refute this conclusion in a study with longer learning time.”
Reviewer 2 Report
Comments and Suggestions for Authors
Thank you for having addressed most of my concerns. The paper is much more readable now and the sophisticated methodology is explained also more clearly. There are still some minor issues, which I list below. Please consider them before resubmitting the final version of your paper.
· Page 5, 3rd paragraph: is the wording “MIDI piano performance” a good description of what is meant? Is MIDI not the tool to record the playing rather than being a performance?
· Page 6, 1st sentence: the wording “developing learning” seems to lack an additional substantive. Learning of “what”? Is ”learning process” not a better option?
· Page 6, line 7: the limitations of the study should be grouped in the discussions section.
· Page 6, 2nd par.: “the same notes were repeatedly played…”. Here the concrete task is mentioned (playing notes) without explaining it. Please try to supply the needed information at first introduction of the experimental design.
· Page 7, Motor sequence task: “an open source…” instead of “a open soure..”?
· Page 9, Digit span: what does the F, S or S stand for in DSP-F, DSP-R, DSP-S? The assignment of the letters seems to be somewhat arbitrary. This is given in the figure caption of figure 3, but it should be given also in the main text.
· Page 10, 2.6. Statistics, line 7: “using” instead of “useing”?
· Page 12, Figure caption of figure 3: what does ZVT stand for? The abbreviation is not explained.
· Page 17,las paragraph: use psychomotor consistently with lower case.
